# Optimizing Medium Composition and Environmental Culture Condition Enhances Antioxidant Enzymes, Recovers *Gypsophila paniculata* L. Hyperhydric Shoots and Improves Rooting *In Vitro*

**DOI:** 10.3390/plants12020306

**Published:** 2023-01-09

**Authors:** Suzan M. Mohamed, Mohammed E. El-Mahrouk, Antar N. El-Banna, Yaser M. Hafez, Hassan El-Ramady, Neama Abdalla, Judit Dobránszki

**Affiliations:** 1Physiology & Breeding of Horticultural Crops Laboratory, Department of Horticulture, Faculty of Agriculture, Kafrelsheikh University, Kafr El-Sheikh 33516, Egypt; 2Genetics Department, Faculty of Agriculture, Kafrelsheikh University, Kafr El-Sheikh 33516, Egypt; 3EPCRS Excellence Center, Plant Pathology and Biotechnology Laboratory, Agricultural Botany Department, Faculty of Agriculture, Kafrelsheikh University, Kafr El-Sheikh 33516, Egypt; 4Soil and Water Department, Faculty of Agriculture, Kafrelsheikh University, Kafr El-Sheikh 33516, Egypt; 5Plant Biotechnology Department, Biotechnology Research Institute, National Research Centre, 33 El Buhouth St., Dokki, Giza 12622, Egypt; 6Centre for Agricultural Genomics and Biotechnology, FAFSEM, University of Debrecen, 4400 Nyíregyháza, Hungary

**Keywords:** paclobutrazol, gas exchange rates, light density, catalase, peroxidase, polyphenol oxidase, electrolyte leakage

## Abstract

*Gypsophila paniculata* L. is one of the most important commercial cut flowers worldwide. The plant is sterile and propagated mainly by *in vitro* culture techniques. However, hyperhydricity hinders its micropropagation and increases mortality during *ex vitro* acclimatization. Hyperhydric shoots of *G. paniculata* were proliferated from nodal explants on MS medium without growth regulators that contained 30 g L^−1^ sucrose, and gelled with 6.0 g L^−1^ agar. Medium components and environmental culture conditions were optimized to revert hyperhydricity in *G. paniculata* microshoots and develop an efficient micropropagation protocol for commercial production. Multiple shoots with high quality were successfully regenerated on MS medium fortified with potassium and ammonium nitrate at full concentration, 2.0 mg L^−1^ paclobutrazol, solidified with 9.0 g L^−1^agar in Magenta boxes of 62.87 gas exchange/day and incubated under light density of 60 µmol m^−2^s^−1^. We recorded 4.33 shoots, 40.00 leaves, 6.33 cm, 2.50 g and 95.00% for number of shoots/explant, number of leaves/shoot, shoot length, shoot fresh weight and normal shoots percentage, respectively. Well-rooted plantlets of *G. paniculata* were developed from the reverted microshoots, with the rooting percentage (95.00%) on MS medium augmented with 1.0 mg L^−1^ IBA in Magenta boxes of 62.87 gas exchange/day and 60 µmol m^−2^s^−1^ light density. *In vitro*-rooted plantlets exhibited reduced electrolyte leakage, and enhanced antioxidant enzymes activity of peroxidase, catalase, and polyphenol oxidase due to good ventilation at the highest gas exchange rate of the culture vessels.

## 1. Introduction

*Gypsophila paniculata* L. (Caryophyllaceae), known as baby’s breath, panicled baby’s-breath or common gypsophila, is a perennial plant but often cultivated commercially as an annual flowering crop. It originated from the temperate region of Europe and Asia [1]. It can be grown as a garden plant and is utilized as one of the most valuable cut flowers as a filler or bedding plant in flower bouquets [2,3]. *G. paniculata* is a major commercial crop in the flower market worldwide. It can also be employed for industrial purposes for making detergents due to its high content of saponin (crude saponin, called saponin album), which is obtained from *G. paniculata* roots and rhizomes. These saponins could be used efficiently against some nematodes [4]. Generally, *G. paniculata* flowers are sterile and do not form seeds. Therefore, stem cuttings are used for its conventional vegetative propagation. However, the low rooting capacity of its terminal cuttings [2] hinders its propagation. Therefore, plant tissue culture is employed for the commercial propagation of *G. paniculata* [5]. Micropropagation presents an efficient and economical method for mass propagation of pathogen-free, genetically uniform plants for the commercial cultivation of the *G. paniculata* plant. However, the occurrence of hyperhydricity during the *in vitro* propagation of *G. paniculata* is a major limitation for its propagation due to the very low survival rate during acclimatization [1,5,6].

Hyperhydricity (HH) is a physiological disorder and morphological malformation caused by unfavorable *in vitro* conditions such as increased relative humidity in culture vessels, the level of sucrose in the medium, low light intensity, high temperature and high cytokinin concentrations [7,8,9,10,11]. Furthermore, high salt concentration of the medium, improper gelling agents and their concentration, and imbalanced mineral concentrations can also lead to HH. Additionally, inefficient gas exchange enhances the accumulation of ethylene in the culture vessels, leading to HH [12,13,14,15]. The regulation of *in vitro* culture conditions favors high quality plantlets and improves their *ex vitro* growth and survival during acclimatization [16]. Hyperhydric plantlets are characterized by their translucent, thick, brittle, wrinkled leaves and high-water content in their tissues. These hyperhydric cultures show physiological malformation and malfunctions leading to low regeneration rate, decreased quality, altered metabolism and inability to survive, which restrains their commercial micropropagation [11,17,18]. There are two strategies that could be applied to control hyperhydricity: preventing the incidence of hyperhydricity, and the reversion of hyperhydric plantlets to their normal appearance. The majority of the previous literature focused on the prevention the occurrence of hyperhydricity in advance, while a few studies reported the reversion of hyperhydric shoots to their normal growth in order to improve the performance of *in vitro* propagation of some species, including the banana plant [19]; *Agave sisalana* Perr. [20]; and *Dianthus chinensis* L. [17,21]. 

The optimization of mineral nutrients in the culture medium is considered as an important mean for improving the micropropagation protocol of plants. Mineral nutrients greatly affect the morphological and physiological growth and development [22]. Nitrogen (N) could be found in the MS medium in two sources, ammonium (NH_4_^+^) and nitrate (NO_3_^−^). Combining the two sources of nitrogen in MS medium rather than using a single nitrogen source only proved effective for plant micropropagation, confirming the key roles of both NO_3_^−^ and NH_4_^+^ in shoot multiplication and hyperhydricity [23]. Moreover, there is a strong relationship between agar concentration in the culture medium and the occurrence of hyperhydricity in tissue cultured plants [1,5,24]. Different concentrations of agar (6, 8, 10 and 12 g L^−1^) were added to the shoot regeneration medium using shoot tip explants. The lowest hyperhydricity percentage (3%) has been noticed when agar was supplied to the culture medium at 12 g L^−1^, while adding 6 g L^−1^ recorded the highest hyperhydricity percentage (17%) in *G. paniculata in vitro* shoots [5]. The addition of gelling agents at optimal concentration to the culture medium can circumvent the abnormal morphogenesis of *in vitro* propagated plants as hyperhydricity [25]. In this concern, among different agar concentration at 0, 4, 6, 8, and 10 g L^−1^, the addition of 10 g L^−1^ agar to the culture medium reduced hyperhydricity to a minimum value (8%), but decreased the number of proliferated shoots from nodal explants of *G. paniculata* [1]. 

Paclobutrazol (PBZ), commonly used as a fungicide, is a member of the triazole family. Triazoles have plant growth regulating potential by inhibiting the biosynthesis of gibberellin [26]. Moreover, triazoles enhance the tolerance of plants to abiotic stresses, such as the oxidative stress caused by hyperhydricity; therefore, they have been recognized as multi-protectants [27]. 

The growth rate and other morphological and physiological features of *in vitro* raised plantlets are influenced by the physical and chemical environments of culture vessels [28]. The sealing type of culture vessels and gas exchange rate could prevent or reverse the physiological disorder of hyperhydricity [29]. Light acts as an energy source for photosynthesis and regulates signals in the biological mechanisms. Conversely, high light intensities could lead to hyperhydricity in tissue cultured plants [10]. Oxidative stress has been reported in hyperhydric tissues of different plant species as *Euphorbia milli* [30]; *Mammillaria gracilis* [31]; garlic [32]; and *Dianthus caryophyllus* [33]. The antioxidative condition of the tissues should be taken into consideration when describing the hyperhydricity [30,34]. The activities of antioxidant enzymes such as peroxidase, polyphenol oxidase, catalase, superoxide dismutase, hydrogen peroxide, and glutathione reductase have been largely linked to the hyperhydricity phenomenon as biomarkers of oxidative stress [30,35].

Given the importance of the *G. paniculata* plant in the global flower market, which is mainly propagated by tissue culture techniques for commercial production, there are many examples in the literature concerning *G. paniculata in vitro* cultures, e.g., [1,2,5,36,37,38,39,40,41,42]. However, unfortunately, none of these reports discussed the recovery of *G. paniculata* hyperhydric shoots to their normal growth. Therefore, the current investigation aimed to revert the hyperhydric *G. paniculata* shoot cultures with the aim of improving large-scale *in vitro* production of this economically important plant. 

## 2. Materials and Methods

This study was conducted during 2020 and 2021 at the Physiology and Breeding of Horticultural Crops Laboratory, Department of Horticulture, Faculty of Agriculture, Kafrelsheikh University, Egypt.

### 2.1. Plant Materials and Establishment of Aseptic Cultures

*G. paniculata* ‘Snowflake’ was cultivated by stem cuttings in a plastic greenhouse. The average daily photosynthetic photon flux density (PPFD) was 1000 µmol m^−2^ s^−1^, the average relative humidity was 70% and the average daily temperature was 25 °C at the nursery of El Kenana Company, Tanta, Egypt. The irrigation occurred when the plants needed it, and fertilization was carried out with irrigation according to the gypsophila fertilization program [43]. *G. paniculata* shoots (8–12 cm long) were excised from one year old plants. After that, they were washed using running tap water for five minutes. Then, they were rinsed two times with distilled water, then cut into nodal explants (3–4 cm) containing one to two nodes. Under sterile conditions, the explants were surface disinfected using ethanol 70% for 1 min then by dipping in 0.1% (*w*/*v*) mercuric chloride containing two or three drops of Tween-20 for 12 min (Loba Chemical Company, Mumbai, India). After three times, they were rinsed with sterile distilled water, and the dead ends of the disinfected nodal explants were discarded. The explants were cultured for 6 weeks in cylindrical glass jar of 375 mL capacity containing 50 mL full MS medium free of plant growth regulators, 3% (*w*/*v*) sucrose and 6.0 g L^−1^ agar for axillary shoots induction. Cultures were incubated at 25 ± 1 °C under a photoperiod of 16 h by cool-white fluorescent light at 30 μmol m^−2^ s^−1^ PPFD. *In vitro* axillary shoots of *G. paniculata* (2.5–3.0 cm long) were separated and cultured in the same culture conditions mentioned above (same jar, medium and incubation conditions). After 4 weeks, the hyperhydric *in vitro* shoots (more than 95% of the proliferated axillary shoots were hyperhydric) were used as explant materials in four experiments, as follows.

#### 2.1.1. Effect of Nitrate Salt Concentrations on Recovery of Hyperhydric Shoots 

*G. paniculata* hyperhydric shoots were cultured onto MS basal medium in the 375 mL jars containing 50 mL medium, 3% (*w*/*v*) sucrose, but it was solidified with 9 g L^−1^ agar. The nitrogen content in the MS medium was modified by adding potassium nitrate and ammonium nitrate salts at different concentrations (Table 1). The number of shoots/explant, shoot length (cm), shoot fresh weight (g), number of leaves/shoot, and normal shoots percentage were estimated after four weeks of culture.

#### 2.1.2. Effect of Agar Concentrations on Recovery of Hyperhydric Shoots

*G. paniculata* hyperhydric shoots were cultured onto MS basal medium in the 375 mL jars containing 50 mL culture medium supplemented with the best two combinations of (KNO_3_ + NH_4_NO_3_) (i.e., 950 + 825 and 1900 + 1650 mg L^−1^), and solidified with five different concentrations of agar (Sigma Chemical Company, Missouri, USA); (7, 8, 9, 10 and 11 g L^−1^). After four weeks, the number of shoots/explant, number of leaves/shoot, shoot length (cm), shoot fresh weight (g), and normal shoots percentage were measured.

#### 2.1.3. Effect of Paclobutrazol on Recovery of Hyperhydric Shoots

*G. paniculata* hyperhydric shoots were cultured on MS basal medium in the 375 mL jars containing 50 mL medium + 3% (*w*/*v*) sucrose and fortified with the two optimal combinations of KNO_3_ and NH_4_NO_3_ (i.e., 950 + 825 and 1900 + 1650 mg L^−1^) and solidified with the optimal concentration of agar (9.0 g L^−1^) according to the obtained results from the first and the second experiments, and augmented with different concentrations of paclobutrazol (Titan Biotech LTD, Rajasthan, India); (0.0, 1.0, 1.5, 2.0, 2.5 and 3.0 mg L^−1^). The number of shoots/explant, shoot fresh weight (g), shoot length (cm), number of leaves/shoot, and normal shoots percentage were measured after four weeks.

#### 2.1.4. Effects of Gas Exchange Rate and Light Density on Recovery of Hyperhydric Shoots

*G. paniculata* hyperhydric shoots were cultured onto MS basal medium with 3% (*w*/*v*) sucrose and fortified with the two optimal combinations of KNO_3_ and NH_4_NO_3_ (i.e., 950 + 825 and 1900 + 1650 mg L^−1^) in 540 mL Magenta boxes of four different filters containing 80 mL medium (Sac O_2_ company, Belgium, Netherlands). The filters had four different Gas Exchange Rates; GER1, 2, 3 and 4 (7.44, 9.14, 10.83, 62.87 GE/day, respectively). Media were fortified with 2 mg L^−1^ paclobutrazol and solidified with 9 g L^−1^ agar. Cultures were kept at 25 ± 1 °C in a culture room with a 16 h photoperiod at two photosynthetic photon flux (PPF) density (30 and 60 µmol m^−2^s^−1^). The shoot length (cm), number of shoots/explant, number of leaves/shoot, shoot fresh weight (g), and percentage of normal shoots were measured after four weeks. 

#### 2.1.5. Culture Conditions 

The pH of tested media was adjusted using 0.1 N KOH and/or HCl to 5.8. The medium was autoclaved for 15 min at 121 °C and 1.2 kg cm^−2^. All cultures in the first, the second and the third experiments were kept at 25 ± 1 °C in a culture room with 16 h photoperiod and PPF of 30 µmol m^−2^s^−1^ light density by cool-white fluorescent lamps. 

### 2.2. In Vitro Rooting of the Hyperhydricity Recovered Shoots

Hyperhydricity recovered shoots (>3 cm long) were cultured in Magenta boxes (containing 80 mL of MS medium) at four different gas exchange rates (7.44, 9.14, 10.83, 62.87 GE/day) and supplemented with 0.0, 0.5 and 1.0 mg L^−1^ IBA for root induction. The culture was incubated at 25 ± 2 °C under a 16 h photoperiod using cool-white fluorescent tubes at PPFD of 60 µmol m^–2^·s^–1^. After 4 weeks of culture, plantlet fresh weight (g), plantlet length (cm), number of leaves, and rooting percentage were recorded. 

#### 2.2.1. Antioxidant Enzymes Activity in *In Vitro* Rooted Plantlets

Three antioxidant enzymes were selected in the current study, i.e., peroxidase (POX), catalase (CAT), and polyphenol oxidase (PPO). The samples taken from *in vitro* rooted plantlets were the fully expanded leaves, and were measured according to Aebi [44] for the activity of CAT, to Hammerschmidt et al. [45] for POX, and to Malik and Singh [46] for PPO.

#### 2.2.2. Electrolyte Leakage in *In Vitro* Rooted Plantlets 

Electrolyte leakage (EL) was measured in leaf discs of *in vitro* rooted plantlets according to Whitlow et al. [47] and Szalai et al. [48], which was modified according to Dewir et al. [49]. Leaf discs of *in vitro* plantlets of different rooting treatments were placed individually into 25 mL deionized water (Milli-Q 50, Millipore, Bedford, MA, USA). Initial electrical conductivity measurements were recorded for each vial using an Acromet AR20 electrical conductivity meter (Fisher Scientific, Chicago, Il., USA). Flasks were then immersed in a hot water bath (Fisher Isotemp, Indiana, PA, USA) at 80 °C for 1 h to induce cell rupture. The vials were again placed on the Innova 2100 platform shaker for 20 h at 21 °C. Final conductivity was measured for each flask. Electrolyte leakage percentage was calculated as (initial conductivity/final conductivity) × 100.
EL (%) = (initial electrical conductivity/final electrical conductivity) × 100
where initial value means the first reading of EC after being shaken for 20 h at room temperature; final value means the second reading of EC after heating at 80 °C for 1 h then shaking for 20 h at 21 °C 

### 2.3. Acclimatization of In Vitro Rooted Plantlets 

*In vitro* plantlets (5–7 cm length, with 4–5 roots) were gently removed from the medium, rinsed well using tap water, then dipped for few seconds in a solution of fungicide (0.5 g L^−1^ Rizolex-T 50% WP that contains 20% *w*/*w* Tolclophos-methyl and 30% *w*/*w* thiuram, Kafr El-Zayat Company, El-Gharbia, Egypt). Finally, they were transplanted into small plastic pots (coffee cups, 8 cm diameter) filled with a sterile mixture of vermiculite and peat moss (1:1; *v*/*v*). The plantlets were covered with a clear coffee cup for the first 20 days of culture in the greenhouse. The environmental condition in the shaded greenhouse was 25 ± 2 °C air temperature, relative humidity (60–70%), and 100 µmol m^−2^ s^−1^ PPF. The plantlets were regularly fertigated using a fertilizer solution (N:P:K at 19:19:19) (Rosasol; Rosier, Moustier, Belgium) at 0.5 g L^−1^ once after 30 days. 

### 2.4. Statistical Analyses

The first experiment was arranged in one factor completely randomized design (nitrate salts concentration in MS medium) and it was statistically analyzed by one-way ANOVA. The second experiment (2 concentrations of nitrate salts × 5 concentrations of agar) and the third experiment (2 concentrations of nitrate salts × 6 concentrations of paclobutrazol) were designed in two factors and analyzed by two-way ANOVA. The fourth experiment was reported in three factors (2 nitrate concentrations × 4 gas exchange rates × 2 light densities). The *in vitro* rooting experiment was reported in two factors (4 gas exchange rates × 3 concentrations of IBA). Each treatment was represented by three replicates. Each replicate was a cylindrical glass jar (in the first three experiments) or Magenta box (in the fourth experiment and the rooting experiment) containing three explants. The data were subjected to ANOVA using Costat statistical analysis software (version 6.311, [50]). The mean separations were calculated using Duncan’s multiple range test [51] at *p* ≤ 0.05 significance level. 

## 3. Results 

### 3.1. Influence of Nitrate Salts on Growth and Hyperhydricity Reversion of G. paniculata In Vitro Shoots 

The concentrations of nitrate salts displayed significant differences for growth parameters and percentage of normal shoots of *G. paniculata in vitro* hyperhydric shoot cultures (Table 2). Decreasing the concentration of both KNO_3_ and NH_4_NO_3_ in MS medium to half strength (950 and 825 mg L^−1^, respectively) produced a significant result of the highest numbers of shoots/explant and leaves/shoot (2.70, 18.00), respectively. Unmodified MS medium resulted in the maximum values of shoot length, shoot fresh weight and percentage of normal shoots (6.70 cm, 0.99 g and 85.00%, respectively) but without significant differences with half nitrates MS medium for shoot length and shoot fresh weight only. The obtained results indicated that half salt strength of potassium and ammonium nitrates produced significantly maximal lateral shoot multiplication of *G. paniculata* as compared with the control treatment. However, the unmodified full nitrates MS medium was optimal for shoot elongation and quality. 

### 3.2. Influence of Agar and Nitrate Salts on Growth and Hyperhydricity Reversion of G. paniculata In Vitro Shoots

Significant differences among the treatments of agar and nitrates for the shoots’ growth parameters of *in vitro*-derived shoots of *G. paniculata*, as well as the percentage of normal shoots, were observed (Table 3). MS medium supplemented with half concentration of both KNO_3_ and NH_4_NO_3_ (950 + 825) and solidified with 9 g L^−1^ agar significantly improved the shoot length (9.00 cm) and enhanced the reversion of hyperhydric shoots to a normal state (95.00%). It is optimal for shoot elongation and shoots quality, as well. There was no significant difference between full- and half-concentration of nitrate with regard to the percentage of normal shoots. However, MS medium unmodified in nitrate salts (1900 + 1650) and solidified with 9 g L^−1^ agar gave the best significant results for the number of shoots/explant, shoots fresh weight and number of leaves/shoot. Thus, modified and unmodified MS in nitrates and solidified with 9 g L^−1^ agar was optimal for lateral shoot multiplication, with no significant differences between them. Increasing agar concentration in culture medium from 7 to 9 g L^−1^ significantly improved shoot elongation and quality of *G. paniculata*, when MS medium was supplemented with half concentration of nitrate salts while, it enhanced shoot growth and multiplication of *G. paniculata* significantly when it was added to unmodified MS medium. Increasing agar concentration in the culture medium >9 g L^−1^ decreased the number of proliferated shoots. 

### 3.3. Influence of Paclobutrazol and Nitrate Salts on the Growth and Hyperhydricity Reversion of G. paniculata In Vitro Shoots

Significant differences among treatments of paclobutrazol and nitrate salts for all recorded parameters were detected (Table 4). The number of leaves per shoot and number of shoots per explant were significantly enhanced at 2 mg L^−1^ paclobutrazol and half concentration of nitrate salts, recording 4.00 shoots and 25.00 leaves per shoot, respectively. However, a maximum percentage of normal shoots (100%), which is about three times more than in the control (0 mg L^−1^ PBZ, 35.00%), was recorded at 2 mg L^−1^ paclobutrazol and the full concentration of nitrate salts. The highest shoot fresh weight (2.10 g) was recorded at 2.5 mg L^−1^ and 3 mg L^−1^ paclobutrazol, and half the concentration of nitrate salts. The addition of 2 mg L^−1^ paclobutrazol to MS medium supplemented with a full concentration of nitrate salts decreased shoot length, but the number of normal shoots were increased as compared with the same medium without paclobutrazol. 

### 3.4. Influence of Gas Exchange Rate, Nitrate Salts and Light Density on Growth and Hyperhydricity Reversion of G. paniculata In Vitro Shoots 

There were significant differences among the treatments of gas exchange rate, nitrate salts concentration in MS medium and light densities for all recorded measurements (Table 5). The tabulated data clearly showed that the highest gas exchange rate (62.87 GE/day) and full concentration of nitrate salts under light density of 60 µmol m^−2^s^−1^ significantly enhanced *in vitro* shoot multiplication of *G. paniculata*, represented in the number of shoots/explant, number of leaves/shoot, shoot length, and shoot fresh weight (4.33 shoots, 40.00 leaves, 6.33 cm, and 2.50 g, respectively), as well as promoting the shoots’ quality by increasing the normal shoots percentage significantly (95.00%). No significant differences were observed between a full and half concentration of nitrate salts in MS medium at (62.87 GE/day) and 60 µmol m^−2^s^−1^ light density with regard to shoot length and normal shoots percentage. 

### 3.5. Influence of Gas Exchange Rate and Indole Butyric Acid on In Vitro Rooting of G. paniculata 

There were significant differences among the treatments of gas exchange rates and indole butyric acid on *in vitro* rooting of *G. paniculata* shoots recovered from hyperhydricity, for all recorded measurements (Table 6). The obtained results showed that the gas exchange rate (62.87 GE/day) and 0.5 mg L^−1^ IBA increased plantlet length and number of leaves significantly (8.70 cm, and 30.00, respectively). However, the plantlet fresh weight was maximum (4.27 g) at 0.5 mg L^−1^ IBA and 7.44 GE/day. The highest gas exchange rate (62.87 GE/day) and 1.0 mg L^−1^ IBA recorded the optimum value for rooting percentage (95.00%). 

### 3.6. Antioxidant Enzymes Activity and Electrolyte Leakage in In Vitro Rooted Plantlets of G. paniculata

The activity of antioxidant enzymes (POX, CAT and PPO) was measured in *in vitro* rooted plantlets of *G. paniculata* (Figure 1). The effect of the gas exchange rate of Magenta box and IBA concentration in the rooting medium on the previous antioxidant enzymes were examined. A gas exchange rate of 10.83 and 62.87 GE/day showed a significant increase in CAT activity at different concentrations of IBA (Figure 1A). On the other hand, CAT activity was lowered at 7.44 and 9.14 GE/day under different concentrations of IBA. However, there was a significant increase in the activity of POX at the lowest gas exchange rate (7.44), compared to the other GER (Figure 1B). Similarly, PPO activity was also increased significantly at 7.44 and 9.14 GE/day for different concentrations of IBA (Figure 1C). The gas exchange rate at 10.83 GE/day resulted the highest decrease of electrolyte leakage (EL) as compared with other gas exchange rates (Figure 1D). However, the lowest gas exchange rate conferred the highest increase in EL. 

### 3.7. Acclimatization of In Vitro Rooted Plantlets 

All *in vitro* rooted plantlets produced from the rooting experiment (Table 6) were acclimatized as described in the materials and methods, and the reverted plantlets were acclimatized with a 100% survival rate.

## 4. Discussion

The commercial micropropagation industry of *G. paniculata* is hindered by hyperhydricity [6]. The main factors influencing the hyperhydricity of *in vitro* propagated plants are the minerals, the hormonal composition of the medium and culture conditions [52]. The optimization of the medium chemical components and the regulation of the environmental culture conditions could be useful to overcome hyperhydricity in tissue cultured plants. 

For improving the micropropagation protocol of a plant species, the concentration of the two nitrogen salts in MS medium (NH_4_NO_3_ and KNO_3_) should be modified and optimized according to the plant species. It would be favorable and more practical than that of separated ions [52]. Increasing N improved shoot elongation and quality, but reduced the number of shoots in most red raspberry cultivars (*Rubus idaeus* L.) [53]. This result was in agreement with that obtained from the current study. 

Hyperhydricity could be reduced by gelling agent supplement to the culture medium at high concentration or with high gel strength [54]. Agar is the most commonly applied gelling agent in the media used for plant tissue culture [1]. Agar was proven to affect both the multiplication rate and quality of the shoots in *Scrophularia yoshimurae* Yamazaki [28]. The propagation rate was decreased at high gelling agent concentration, due to the decreased availability of water and dissolved nutrients [25]. Conversely, the previous studies reported that increasing agar concentration to an optimal level could overcome hyperhydricity of *G. paniculata* [1] and *Anthyllis barba-jovis* L. [55]. 

Paclobutrazol is a plant growth retardant, and an inhibitor of gibberellin biosynthesis. It is widely applied to culture medium to retard shoot elongation as well as hyperhydricity in liquid cultures, and to enhance the growth, development and quality of *in vitro* cultured plants [56,57]. Inhibiting cell division and elongation, increasing chlorophyll content and enhancing protein accumulation, increasing absorption of mineral nutrients, reducing water consumption, increasing photosynthetic capacity, and promoting tolerance to environmental stresses are all thought to be part of PBZ’s mechanism for combating hyperhydricity [58,59,60]. Paclobutrazol at 2 mg L^−1^ had inhibited the adverse effect of drought stress that was stimulated by polyethylene glycol (PEG) on photosynthetic pigments in *Stevia rebaudiana* Bertoni *in vitro* cultures [27]. In the current study, paclobutrazol negatively affected the number of shoots/explant, shoot fresh weight, shoot length, number of leaves/shoot, and normal shoots percentage. In addition, the percentage of normal shoots was reduced from 95% to 25% at half nitrates (950 + 825). However, at optimal conditions of 60 µmol m^−2^ s ^−1^ PPFD and 62.87 gas exchange rate, all measured parameters were significantly improved. Paclobutrazol is considered as an anti-gibberellin, but at optimal light density and gas exchange the photosynthesis and antioxidant activity could be stimulated. The addition of 2 mg L^−1^ PBZ to MS medium supplemented with full nitrate salts and solidified with 9 g L^−1^, at a light density of 60 µmol m^−2^s^−1^ and gas exchange rate (62.87 GE/day), is recommended for optimal shoot growth, multiplication and quality. 

Different *in vitro* methods have been utilized to enhance the photosynthetic rate, growth and quality of micro-propagated plants [61]. Modified bottles or caps which allow gas exchange [62], increasing light intensity [63], or a combination of these methods could be used. Gas exchange boxes reduce the relative humidity in the culture vessels [64], increase gas exchange rate with the outer atmosphere [65] and decrease water availability, providing suitable environmental conditions for photoautotrophic micro-propagated plants [61]. 

Modifying the lid of culture bottles ensured good ventilation, overcame hyperhydricity (0%) and induced normal development of *G. paniculata* node cultures. Culture vessels closed with a polypropylene lid recorded 80% hyperhydric shoots [65]. Another investigation stated that polypropylene caps and two layers of foil were the optimal vessel closures for shoot proliferation of *G. paniculata* ‘Bristol fairy’ shoot cultures [41]. Furthermore, hyperhydricity was reduced significantly in *Scrophularia yoshimurae* Yamazaki shoot cultures when culture vessels were closed by filter paper, while aluminum foil led to a higher number of hyperhydric shoots [28]. Moreover, it was found that the aerated bioreactor decreased hyperhydricity rate to 2.90–3.20% compared to an unaerated system (19.30%), and promoted growth and proliferation of high-quality shoots for the large-scale production of *G. paniculata* [3]. In addition, ventilated cultures of carnations lowered the hyperhydricity percentage (38.4%) when compared with unventilated ones, which recorded 62.7% hyperhydrated shoots [66].

More investigations have been performed to minimize hyperhydricity and enhance shoots proliferation and growth in tissue cultured plants, depending on different or modified culture vessels that allow gas exchange [25,67,68,69]. Therefore, Magenta boxes equipped with gas exchange filters should be utilized as a promising culture environment for improving the micropropagation protocol [70]. From the current study, we deduce that the best treatment for *in vitro* shoot multiplication of *G. paniculata*, and shoot quality as well, was the green filter of the highest gas exchange rate, MS medium containing the full nitrates, and 60 µmol m^−2^s^−1^ light density (Figure 2). A linear increase in rooting ability of most magnolia genotypes, by increasing concentration of IBA in the medium from 1 to 6 mg L^−1^, was detected [71]. These results were in agreement with the obtained results, where there was an increase in the rooting percentage of *G. paniculata* by increasing the concentration of IBA in rooting medium at different gas exchange rates.

Ventilation conditions amended the quality of plantlets by enhancing the activity of antioxidant enzymes and reducing the electrolyte leakage. It was recorded that the most powerful CAT activities were induced under blue LEDs in ventilated cultures of carnations [66]. In the present investigation, the different filters that allow gas exchange improved the intracellular antioxidant enzyme activities of *G. paniculata* (CAT, POX, and PPO). Stimulating the antioxidant enzymes activity could maintain the optimal function of the tissues [72]. It is the defense response that protects the plant cells from reactive oxygen species (ROS), which cause harmful oxidative stress to plant cells [73,74]. Electrolyte leakage has negative effects on cell membranes as a result of different stresses [75]. In this study, an increase in electrolyte leakage was observed at the lowest gas exchange rate due to the low ventilation condition in culture vessels that increased the antioxidant activity dramatically.

## 5. Conclusions

An efficient micropropagation protocol for large scale production has been developed through reverting *G. paniculata* hyperhydric shoots. Some modifications to the medium component and environmental culture conditions were performed to enhance growth, multiplication rate and quality of *G. paniculata in vitro* shoot cultures. Hyperhydric shoots of *G. paniculata* were recovered successfully with high multiplication rate and quality on MS medium supplemented with full concentration of potassium- and ammonium nitrates, with 2 mg L^−1^ PBZ and solidified with 9 g L^−1^ agar in Magenta boxes of gas exchange rate of 62.87 GE/day, under light density 60 µmol m^−2^s^−1^. These healthy, recovered, multiple shoots of *G. paniculata* were rooted efficiently on MS medium supplemented with 1.0 mg L^−1^ IBA and high gas exchange rate of 62.87 GE/day. These conditions facilitated the acclimatization and survival of *G. paniculata* plantlets.

## Figures and Tables

**Figure 1 plants-12-00306-f001:**
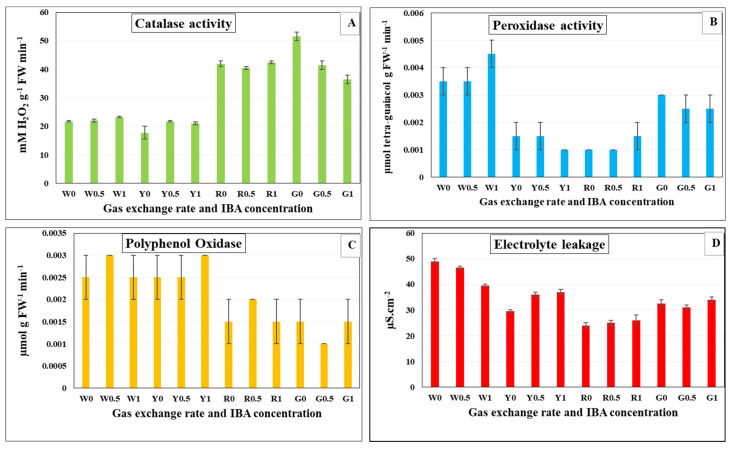
Effect of gas exchange rate and indole butyric acid concentration on antioxidant enzymes activity and electrolyte leakage of *G. paniculata in vitro* rooted plantlets after four weeks of culture. (**A**) Catalase (**B**) Peroxidase (**C**) Polyphenol oxidase, and (**D**) Electrolyte leakage. W0 = 7.44 GE/day at 0 mg L^−1^ IBA, W0.5 = 7.44 GE/day at 0.5 mg L^−1^ IBA, W1 = 7.44 GE/day at 1.0 mg L^−1^ IBA, Y0 = 9.14 GE/day at 0 mg L^−1^ IBA, Y0.5 = 9.14 GE/day at 0.5 mg L^−1^ IBA, Y1 = 9.14 GE/day at 1.0 mg L^−1^ IBA, R0 = 10.83 GE/day at 0 mg L^−1^ IBA, R0.5 = 10.83 GE/day at 0.5 mg L^−1^ IBA, R1 = 10.83 GE/day at 1.0 mg L^−1^ IBA, G0= 62.87 GE/day at 0 mg L^−1^ IBA, G0.5 = 62.87 GE/day at 0.5 mg L^−1^ IBA, G1 = 62.87 GE/day at 1.0 mg L^−1^ IBA. (W: White filter 7.44; Y: Yellow filter 9.14; R: Rose filter 10.83 and G: Green filter 62.87 GE/day).

**Figure 2 plants-12-00306-f002:**
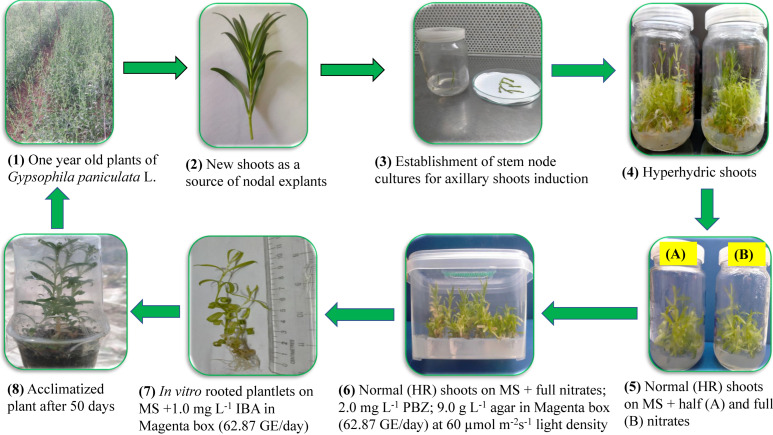
A novel protocol for producing healthy, non-hyperhydric *in vitro* plantlets of *G. paniculata* by recovering from hyperhydrated *in vitro* shoots, where HR is hyperhydricity recovered.

**Table 1 plants-12-00306-t001:** Potassium and ammonium nitrate salts concentration in MS medium.

Treatments	Supplemented Salts (mg L^−1^)		Final Concentrations of K^+^, NO_3_^−^, and NH_4_^+^ Have Been Calculated in mg L^−1^				
	KNO_3_	NH_4_NO_3_	K^+^ (KNO_3_)	NO_3_^−^ (KNO_3_)	NH_4_^+^ (NH_4_NO_3_)	NO_3_^−^ (NH_4_NO_3_)	Total NO_3_^−^
# Full nitrates MS	1900	1650	730	1165	371	1278	2443
½ nitrates MS	950	825	367	582	185	639	1803
¼ nitrates MS	470	413	181	288	92.9	320	608
^1/8^ nitrates MS	235	206	90.9	144	46.4	159	303
^1/16^ nitrates MS	120	103	46.4	73.6	23.18	79.8	153.4

# Control treatment = unmodified MS medium. Purity of KNO_3_ and NH_4_NO_3_ was 99% (Technogene Corp, Cairo, Egypt). MS medium was supplemented with nitrogen from two sources of nitrate and one source of ammonium in all treatments. Molecular weight of KNO_3_ = 101.1; NH_4_NO_3_ = 80; K ^+^ = 39.1; NO_3_^−^ = 62; NH_4_^+^ = 18 g/mole.

**Table 2 plants-12-00306-t002:** Effect of potassium nitrate (KNO_3_) and ammonium nitrate (NH_4_NO_3_) on growth and hyperhydricity reversion of *G. paniculata in vitro* shoots after four weeks of *in vitro* culture. All media were solidified with 9 g L^−1^ agar. All cultures were kept at a light density of 30 µmol m^−2^s^−1^.

Treatment (mg L^−1^) KNO_3_ + NH_4_NO_3_	Shoot Length (cm)	No. of Shoots/Explant	No. of Leaves/Shoot	Shoot Fresh Weight (g)	Normal Shoots (%)
120 + 103	3.00 d	1.00 b	9.00 c	0.09 d	33.00 d
235 + 206	5.00 bc	1.00 b	11.70 b	0.28 cd	33.00 d
470 + 413	3.83 cd	1.70 b	14.00 b	0.47 bc	50.00 c
950 + 825	5.70 ab	2.70 a	18.00 a	0.76 ab	75.00 b
#1900 + 1650	6.70 a	1.33 b	13.33 b	0.99 a	85.00 a
Significance	**	**	***	***	**

NS, **, *** Non-significant or significant at *p* ≤ 0.05, 0.01, and 0.001, respectively, according to Duncan’s multiple range tests followed by ANOVA. Values followed by the same letters in the same column were not significantly different. # Control treatment (unmodified MS medium).

**Table 3 plants-12-00306-t003:** Effect of agar and nitrate salts on growth and hyperhydricity reversion of *G. paniculata in vitro* shoots after four weeks. All cultures were kept at a light density of 30 µmol m^−2^s^−1^.

Treatments KNO_3_ + NH_4_NO_3_		Shoot Length (cm)	No. of Shoots/Explant	No. of Leaves/Shoot	Shoot Fresh Weight (g)	Normal Shoots (%)
Agar (g L^−1^)	(mg L^−1^)					
7	950 + 825	8.00 b	3.0 de	21.67 ab	1.70 a	29.33 cd
	1900 + 1650	6.67 c	3.0 de	15.00 c	1.30 ab	25.00 d
8	950 + 825	8.30 ab	4.0 bcd	23.30 ab	1.73 a	33.30 c
	1900 + 1650	5.00 e	4.0 bcd	20.00 b	1.30 ab	41.67 c
9	950 + 825	9.00 a	5.0 ab	20.00 b	0.37 c	95.00 a
	1900 + 1650	6.00 cd	6.0 a	25.00 a	1.83 a	90.00 a
10	950 + 825	6.00 cd	4.3 bc	23.30 ab	0.80 bc	50.00 b
	1900 + 1650	6.00 cd	1.3 f	20.00 b	1.60 a	58.00 b
11	950 + 825	5.67 de	3.3 cde	21.67 ab	1.50 a	33.30 c
	1900 + 1650	5.00 e	2.3 e	23.30 ab	1.73 a	41.67c
Significance	N	***	**	Ns	*	Ns
	A	***	***	*	Ns	***
	N × A	***	***	***	***	*

NS, *, **, *** Non-significant or significant at *p* ≤ 0.05, 0.01, and 0.001, respectively, according to Duncan’s multiple range tests followed by ANOVA. Values followed by the same letters in the same column were not significantly different. N and A indicate the significance of the single effect of nitrate salts and agar, respectively. N × A indicates the significance of the interaction between nitrate salts and agar.

**Table 4 plants-12-00306-t004:** Effect of paclobutrazol and nitrate salts on growth and hyperhydricity reversion of *G. paniculata in vitro* shoots after four weeks. All media were solidified with 9 g L^−1^ agar. All cultures were kept at light density of 30 µmol m^−2^s^−1^.

Treatments		Shoot Length (cm)	No. of Shoots/Explant	No. of Leaves/Shoot	Shoot Fresh Weight (g)	Normal Shoots (%)
Paclobutrazol (mg L^−1^)	KNO_3_ + NH_4_NO_3_ (mg L^−1^)					
0	950 + 825	9.00 a	3.67 ab	21.67 bc	1.70 ab	50.00 c
	1900 + 1650	6.00 c	1.33 fg	15.00 d	1.31 bcd	35.00 d
1.0	950 + 825	4.00 d	1.00 g	15.00 d	0.37 e	35.00 d
	1900 + 1650	4.33 d	2.67 cde	25.00 a	1.30 bcd	65.00 c
1.5	950 + 825	6.50 c	1.00 g	15.00 d	0.80 de	25.00 e
	1900 + 1650	3.33 ef	3.67 ab	20.00 c	1.63 ab	50.00 c
2.0	950 + 825	6.00 c	4.00 a	25.00 a	1.50 abc	25.00 e
	1900 + 1650	3.50 efd	2.33 de	20.00 c	1.60 abc	100.00 a
2.5	950 + 825	6.00 b	3.33 abc	23.33 ab	1.03 cd	35.00 d
	1900 + 1650	3.33 ef	3.00 bcd	23.33 ab	2.00 a	25.00 e
3.0	950 + 825	6.00 c	3.00 bcd	21.67 bc	2.10 a	25.00 e
	1900 + 1650	2.67 f	2.00 ef	20.00 c	1.67 ab	25.00 e
Significance	N	***	NS	NS	**	*
	*p*	*	***	***	***	***
	N × *p*	***	***	***	***	**

NS, *, **, *** Non-significant or significant at *p* ≤ 0.05, 0.01, and 0.001, respectively, according to Duncan’s multiple range tests followed by ANOVA. N and *p* indicate the significance of the single effect of nitrate salts and paclobutrazol, respectively. Values followed by the same letters in the same column were not significantly different. N × *p* indicates the significance of the interaction between nitrate salts and paclobutrazol.

**Table 5 plants-12-00306-t005:** Effect of gas exchange rate, nitrate salts and light density on growth and hyperhydricity reversion of *G. paniculata in vitro* shoots after four weeks. All media were fortified with 2 mg L^−1^ paclobutrazol and solidified with 9 g L^−1^ agar.

Treatments			Shoot Length (cm)	No. of Shoots/ Explant	No. of Leaves/ Shoot	Shoot Fresh Weight (g)	Normal Shoots (%)
GER (GE/day)	KNO_3_ + NH_4_NO_3_ (mg L^−1^)	PPFD (µmol m^−2^s^−1^)					
GER1 (7.44)	950 + 825	30	2.00 h	1.00 d	10.00 e	0.90 e	35.00 h
	1900 + 1650	30	3.67 cdef	1.67 cd	13.33 de	1.17 de	25.00 i
	950 + 825	60	2.33 gh	2.00 bcd	15.00 cde	1.00 e	25.00 i
	1900 + 1650	60	3.00 efgh	1.00 d	10.00 e	0.90 e	35.00 h
GER2 (9.14)	950 + 825	30	2.00 h	1.00 d	10.00 e	0.90 e	35.00 h
	1900 + 1650	30	5.00 b	3.00 b	20.00 bc	1.50 cd	35.00 h
	950 + 825	60	2.67 fgh	1.67 cd	13.33 de	0.97 e	50.00 fg
	1900 + 1650	60	4.00 bcde	2.00 bcd	15.00 cde	1.00 e	55.00 f
GER3 (10.83)	950 + 825	30	2.67 fgh	2.00 bcd	23.33 b	1.83 bc	50.00 fg
	1900 + 1650	30	3.33 defg	1.33 cd	12.67 de	1.17 de	66.67 de
	950 + 825	60	5.00 b	2.00 bcd	35.00 a	2.13 ab	70.00 de
	1900 + 1650	60	4.00 bcde	2.00 bcd	15.00 cde	1.13 de	85.00 bc
GER4 (62.87)	950 + 825	30	4.67 bc	2.00 bcd	17.33 cd	1.27 de	67.00 de
	1900 + 1650	30	4.33 bcd	2.33 bc	16.67 cd	1.67 de	75.00 d
	950 + 825	60	7.00 a	2.00 bcd	15.00 cde	1.17 de	90.00 ab
	1900 + 1650	60	6.33 a	4.33 a	40.00 a	2.50 a	95.00 a
Significance	N		**	*	Ns	Ns	Ns
	GER		***	***	***	***	***
	L		***	Ns	***	Ns	***
	N × GER		***	**	***	***	***
	N × L		**	Ns	Ns	Ns	Ns
	GER × L		***	Ns	***	**	***
	N × GER × L		NS	**	***	***	***

NS, *, **, *** Non-significant or significant at *p* ≤ 0.05, 0.01, and 0.001, respectively, according to Duncan’s multiple range tests followed by ANOVA. Values followed by the same letters in the same column were not significantly different. N, GER, and L indicate the significance of the single effect of nitrate salts, gas exchange rate and light density, respectively. N × GER; N × L and GER × L indicate the significance of the dual interaction between two factors. N × GER × L indicates the significance of the triple interaction between three factors (nitrate salts, gas exchange rate and light intensity), where N = Nitrate salts, GER = Gas Exchange Rate, L = light intensity.

**Table 6 plants-12-00306-t006:** Effect of gas exchange rate and indole butyric acid on *in vitro* rooting of *G. paniculata* hyperhydric recovered microshoots after four weeks. All cultures were kept at light density of 60 µmol m^−2^s^−1^.

Treatments		Plantlet Length (cm)	No. of Leaves	Plantlet Fresh Weight (g)	Rooting (%)
GER (GE/day)	IBA (mg L^−1^)				
GER1 (7.44)	0.0	5.30 ef	18.33 bc	2.50 cd	25.00 e
	0.5	8.00 ab	21.67 b	4.27 a	25.00 e
	1.0	5.00 ef	13.33 c	1.47 d	65.00 c
GER2 (9.14)	0.0	6.00 cdef	15.00 bc	2.00 cd	25.00 e
	0.5	6.70 bcde	13.33 c	2.67 c	25.00 e
	1.0	7.30 abcd	16.67 bc	2.03 cd	50.00 d
GER3 (10.83)	0.0	5.70 cde	13.33 c	2.33 cd	65.00 c
	0.5	7.70 a	30.00 a	4.10 ab	65.00 c
	1.0	4.70 f	16.67 bc	2.83 c	85.00 b
GER4 (62.87)	0.0	6.70 bcde	18.33 bc	2.27 cd	50.00 d
	0.5	8.70 a	30.00 a	3.27 abc	65.00 c
	1.0	6.70 bcde	18.33 bc	3.00 bc	95.00 a
Significance	I	*	**	ns	***
	GER	***	***	***	***
	I × GER	*	**	*	***

NS, *, **, *** Non-significant or significant at *p* ≤ 0.05, 0.01, and 0.001, respectively, according to Duncan’s multiple range tests followed by ANOVA. Values followed by the same letters in the same column were not significantly different. I × GER indicates the significance of the interaction between Indole butyric acid and gas exchange rate of Magenta box, where I = Indole butyric acid and GER = Gas Exchange Rate.

## Data Availability

Not applicable.

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
