# Peer review of "Optimizing Medium Composition and Environmental Culture Condition Enhances Antioxidant Enzymes, Recovers Gypsophila paniculata L. Hyperhydric Shoots and Improves Rooting In Vitro"

_plants, 2023, doi:10.3390/plants12020306_

Round 1

Reviewer 1 Report

The article can be published after minor revision.

Author Response

Reviewer 1#

Response: Your comments have been followed in the revised MS, thanks

Response: The medium has been autoclaved for 15 min at 121ºC.  It is the suitable or temperature used for sterilization of the medium to kill most of microbes  

Reviewer 2 Report

General remark: the work brings valuable results, but it needs to be changed a lot and the English expression needs to be improved.

Many times things are repeated (in materials and methods), and the English language is bad. The work must be sent to professional English editing.

 The title is quite long, and if we ignore that, it should follow the order of the results in the paper (quality of the shoots, rooting, antioxidant enzymes)

64-70: same things are repeated in these two sentences: high cytokinin concentrations  …. high growth regulator levels.

87: unusual English language expression

102: „Paclobutrazol (PBZ), a member of triazole family which is commonly used as fungicides.“ –unfinished sentence

102-105: Unclear, what did you want to say writing about Paclobutrazol?

146-152: The cultivation conditions are identical and needlessly repeated. It would be better to rewrite these two sentences in such a way that the conditions are not repeated.

156-158: Type of vessels…find a way to describe it once

161: After how many days/weeks did you estimate the parameters?

168-169 and 176-177: again, you are repeating which kind of vessels you used 

196-197: “…covered with four different filters of different rates of gas exchange at two light densities 30 and 60 µmol m-2s -1 .”  - this was already said a few lines before

198: I suggest that you summarize the above conditions (volume of vessels, light...) under this subtitle Culture conditions; otherwise it is too repetitive

2.2.2. It is necessary to briefly describe the methodology for measuring electrical conductivity. Without a description, it is unclear what the initial value means and what the final value represents.

238: You cannot start this sentence with “while” – English language

262: You cannot use “whereas” at the beginning of this sentence. “However” would be appropriate. – English language

Table 2: How is it possible that in the initial phase of in vitro culture for the purpose of multiplying a sufficient number of shoots with full strength MS, 3% of sucrose and 6% agar you have 95% of hyperhydric shoots, and in identical conditions in the experiment "Effect of nitrate salt concentrations on recovery of hyperhydric shoots" 85% of shoots were normal?

243-245: So, you have 9 shoots per treatment. That is a too low number, especially for estimation percentage of normal shoots.

278: Inappropriate use of “whereas”

352: Inappropriate use of “while”

Although you described the acclimatization in the materials and methods, you did not mention the results of the acclimatization. Please supplement the results with those data.

402: “It is the most commonly…” – what is the most commonly applied gelling agent?

Fig. 2. Photos are of low quality

Author Response

Reviewer 2#

Comments and Suggestions for Authors

General remark: the work brings valuable results, but it needs to be changed a lot and the English expression needs to be improved.

Response: Many thanks for your comments and your encouragement! We followed your comments to improve the revised MS, thanks again!

Many times things are repeated (in materials and methods), and the English language is bad. The work must be sent to professional English editing.

Response: done, the revised MS was edited and English language improved, Thanks  

The title is quite long, and if we ignore that, it should follow the order of the results in the paper (quality of the shoots, rooting, antioxidant enzymes)

Response: We mean from the title that optimal medium composition and the suitable culture condition improved antioxidant enzymes activity that enhances tolerance to the oxidative stress (resulted by the excess water in hyperhydric shoots), thus the hyperhydric shoots recovered to normal ones, consequence improve rooting. This is the logic order or sequence of events happened. That is why we didn’t follow the order of the results.

64-70: same things are repeated in these two sentences: high cytokinin concentrations…. high growth regulator levels.

Response: Done, (high growth regulator levels) sentence is deleted

87: unusual English language expression

Response: The sentence is corrected

102: „Paclobutrazol (PBZ), a member of triazole family which is commonly used as fungicides“ –unfinished sentence

Response: The sentence is corrected

Paclobutrazol (PBZ), which is commonly used as fungicides, is a member of triazole family. Triazoles have plant growth regulating potential by inhibiting biosynthesis of gibberellin [25]. Moreover, triazoles have been proven to enhance the tolerance of plants to abiotic stresses; such as the oxidative stress caused by hyperhydricity; so they have been recognized as multi-protectants [26].

102-105: Unclear, what did you want to say writing about Paclobutrazol?

Response: We mean paclobutrazol could enhance the tolerance of plants to oxidative stress caused by hyperhydricity due to the excess water in the tissues

146-152: The cultivation conditions are identical and needlessly repeated. It would be better to rewrite these two sentences in such a way that the conditions are not repeated.

Response: Done, the two sentences have been rewritten to avoid the repeated conditions  

156-158: Type of vessels…find a way to describe it once

Response:  Done, thanks

161: After how many days/weeks did you estimate the parameters?

Response:  after four weeks of culture, we wrote that in the title of table (2)

168-169 and 176-177: again, you are repeating which kind of vessels you used

Response:  Done, thanks

196-197: “…covered with four different filters of different rates of gas exchange at two light densities 30 and 60 µmol m-2-1. this was already said a few lines before

Response:  Done, the repeated sentence was deleted, thanks

198: I suggest that you summarize the above conditions (volume of vessels, light...) under this subtitle Culture conditions; otherwise it is too repetitive

Ok, done

2.2.2. It is necessary to briefly describe the methodology for measuring electrical conductivity. Without a description, it is unclear what the initial value means and what the final value represents.

Response:  the methodology for measuring electrical conductivity have been described briefly

238: You cannot start this sentence with “while” – English language

Response:  while is deleted

262: You cannot use “whereas” at the beginning of this sentence. “However” would be appropriate. – English language

Response:  Whereas is replaced by However

Table 2: How is it possible that in the initial phase of in vitro culture for the purpose of multiplying a sufficient number of shoots with full strength MS, 3% of sucrose and 6% agar you have 95% of hyperhydric shoots, and in identical conditions in the experiment "Effect of nitrate salt concentrations on recovery of hyperhydric shoots" 85% of shoots were normal?

Response: we checked the MS carefully (original version), agar concentration in the first experiment "Effect of nitrate salt concentrations on recovery of hyperhydric shoots" was 9 g/L not 6 g/L so, the normal shoots improved and in the following experiment (agar + nitrates) we examine lower and higher concentrations around 9 g/L agar  

We corrected it in the revised MS, thanks

243-245: So, you have 9 shoots per treatment. That is a too low number, especially for estimation percentage of normal shoots.

Response: Yes, each treatment was represented by three replicates; each replicate contained three explants (shoots) because we need enough plant materials for 5 experiments, the first (5 treatments of nitrates concentrations); the second (10 treatments=2 concentrations of nitrate salts × 5 concentrations of agar); the third (12 treatments=(2 concentrations of nitrate salts × 6 concentrations of paclobutrazol); the fourth (16 treatments=2 nitrate concentrations × 4 gas exchange rates × 2 light densities), the fifth (12 treatments=4 gas exchange rates × 3 concentrations of IBA) and if we multiply number of treatment by no. of replicates

The first we need 5×3=15 jars, the second 10×3=30 jars, the third 12×3=36 jars and so on and

278: Inappropriate use of “whereas”

Response:  whereas is deleted

352: Inappropriate use of “while”

Response:  while is deleted

Although you described the acclimatization in the materials and methods, you did not mention the results of the acclimatization. Please supplement the results with those data.

Response: All in vitro rooted plantlets, resulted from the micropropagation protocol, were subjected to acclimatization method described in the materials and methods, and all of them were acclimatized successfully by 100%.  Done , we put that in the result section, thanks

402: “It is the most commonly…” – what is the most commonly applied gelling agent?

Response: Agar is the most commonly applied gelling agent in plant tissue culture medium

The sentence is corrected, thanks

Fig. 2. Photos are of low quality

Response: Unfortunately, we don’t have better quality photos than the provided, but we enlarged them to be clearer, we hope that       

Reviewer 3 Report

The manuscript entitled “Optimizing Medium Composition and Environmental Culture Condition Enhances Antioxidant Enzymes, Recovers Gypsophila paniculata L. Hyperhydric Shoots and Improves Rooting In Vitro” aims to revert hyperhydricity in Gypsophila paniculata L. hyperhydric shoot cultures for improving growth and quality of in vitro propagated plants for a large-scale production of this economically important plant.

The topic is interesting and certain results presented are useful. However, I suggest some improvements should be made.

Тhe full Latin name of the plant is written only when it is first mentioned in the abstract and in the text. After that you should use G. paniculata, not G. paniculata L

The keywords Chlorophyll; Copper; Selenium; Photosynthetic pigments should be deleted, because they are not subject of research in the manuscript.  The authors should replace them with Paclobutrazol; Gas Exchange Rates; Electrolyte Leakage; Light Density.

Please correct the concentration of Paclobutrazol in the Results section. The concentration of paclobutrazol is given in Materials and Methods as mg per liter and in Results as g per liter. According to the data in Table 4, Paclobutrazol has no positive effect on the measured parameters (Number of shoots/explant, shoot fresh weight (g), shoot length (cm), number of leaves/shoot, and normal shoots percentage), especially at 950 + 825 treatment and the percentage of normal shoots was reduced to 25% at optimal concentration of 2 mg L-1 Paclobutrazol. Interestingly, in optimum conditions at light density 60 µmol m-2 s -1 and gas exchange rate 62.87 the all measured parameters are improved, however the nutrient medium containing 2 mg Paclobutrazol.  Please add more details in the Discussion section.  How would you explain that?

If possible provide better quality photos (Figure 2).

The other remarks are given in attached file.

Author Response

Reviewer 3#

Comments and Suggestions for Authors

The manuscript entitled “Optimizing Medium Composition and Environmental Culture Condition Enhances Antioxidant Enzymes, Recovers Gypsophila paniculata L. Hyperhydric Shoots and Improves Rooting In Vitro” aims to revert hyperhydricity in Gypsophila paniculata L. hyperhydric shoot cultures for improving growth and quality of in vitro propagated plants for a large-scale production of this economically important plant.

The topic is interesting and certain results presented are useful. However, I suggest some improvements should be made.

Response: Many thanks for your supporting comments!

Тhe full Latin name of the plant is written only when it is first mentioned in the abstract and in the text. After that you should use G. paniculata, not G. paniculata L

Response: Ok, done, except in the references should be full name, is it right?

The keywords Chlorophyll; Copper; Selenium; Photosynthetic pigments should be deleted, because they are not subject of research in the manuscript.  The authors should replace them with Paclobutrazol; Gas Exchange Rates; Electrolyte Leakage; Light Density.

Response: Ok, done, thanks

Please correct the concentration of Paclobutrazol in the Results section. The concentration of paclobutrazol is given in Materials and Methods as mg per liter and in Results as g per liter. 

Response: Ok, done, thanks

According to the data in Table 4, Paclobutrazol has no positive effect on the measured parameters (Number of shoots/explant, shoot fresh weight (g), shoot length (cm), number of leaves/shoot, and normal shoots percentage), especially at 950 + 825 treatment and the percentage of normal shoots was reduced to 25% at optimal concentration of 2 mg L-1 Paclobutrazol. Interestingly, in optimum conditions at light density 60 µmol m-2 s -1 and gas exchange rate 62.87 the all measured parameters are improved, however the nutrient medium containing 2 mg Paclobutrazol.  Please add more details in the Discussion section.  How would you explain that?

Response:  done, thanks

If possible provide better quality photos (Figure 2).

Response: Unfortunately, we don’t have better quality photos than the provided, but we enlarged them to be clearer, we hope that       

The other remarks are given in attached file.

Response: Done, we followed your comments in the revised MS

Round 2

Reviewer 2 Report

The authors have significantly improved the manuscript, and I think it is worth publishing. However, I would like to point out two things that authors must consider:

1.       If you analyzed all the experiments after 4 weeks, how is it possible that you have such big differences under the same conditions? I'm talking about tables 2 and 3, where in table 2 at 9 g/l agar (you say you corrected the error compared to the original version of the manuscript) and half salt strength of potassium and ammonium nitrates you have a significantly higher multiplication of shoots, and in table 3 . you have a significantly higher multiplication with unmodified MS medium? Thus, the results in text 261-264 are contradictory to the results stated in lines 287-288.

 2.       Subchapter 2.1.5. (Culture conditions) is redundant (except for the first two sentences that you can fit somewhere in the first description of media preparation) because you have described all cultivation conditions in detail in experiments 1-4.

Author Response

Reviewer 2#

Comments and Suggestions for Authors

The authors have significantly improved the manuscript, and I think it is worth publishing. However, I would like to point out two things that authors must consider:

Response: many thanks for your comments, which improved the MS!

  1. If you analyzed all the experiments after 4 weeks, how is it possible that you have such big differences under the same conditions? I'm talking about tables 2 and 3, where in table 2 at 9 g/l agar (you say you corrected the error compared to the original version of the manuscript) and half salt strength of potassium and ammonium nitrates you have a significantly higher multiplication of shoots, and in table 3. you have a significantly higher multiplication with unmodified MS medium? Thus, the results in text 261-264 are contradictory to the results stated in lines 287-288.

Response: many thanks for your comment, yes, sorry for this error!

There is the mistyping in the table 3 for the treatment of modified MS in nitrates and solidified with 9 g L-1 agar (will be 5 ab instead of 5 b). So that, there is no significance differences between modified and unmodified MS in nitrates for table 2 and 3.

  1. Subchapter 2.1.5. (Culture conditions) is redundant (except for the first two sentences that you can fit somewhere in the first description of media preparation) because you have described all cultivation conditions in detail in experiments 1-4.

Response: many thanks for your comment,

Yes, we corrected the text to be right by removing the repetition, thanks again!

many thanks again

Reviewer 3 Report

Dear Authors and Editor,

I read the revised manuscript, as well as the author response file. I’m satisfied with the corrections. 

I noticed some technical errors.

Page 2 line 44

Write the full botanical name of the plant when first mentioned in the introduction Gypsophila paniculata L.

Page 11 line 391 correct table 6 to Table 6

Page 11 line 431

Correct 2 g L-1PBZ to 2 mg L-1PBZ

Best regards!

Author Response

Reviewer 3#

Comments and Suggestions for Authors

I read the revised manuscript, as well as the author response file. I’m satisfied with the corrections.

I noticed some technical errors.

Response: many thanks for your comments, which improved the MS!

Page 2 line 44

Write the full botanical name of the plant when first mentioned in the introduction Gypsophila paniculata L.

Response: Done, many thanks for your comment!

Page 11 line 391 correct table 6 to Table 6

Response: Done, many thanks for your comment!

Page 11 line 431

Correct 2 g L-1PBZ to 2 mg L-1PBZ

Response: Done, many thanks for your comment!

many thanks again! 
